# In Search of *Apis mellifera pomonella* in Kazakhstan

**DOI:** 10.3390/life13091860

**Published:** 2023-09-03

**Authors:** Kamshat Temirbayeva, Aibyn Torekhanov, Ulzhan Nuralieva, Zhanar Sheralieva, Adam Tofilski

**Affiliations:** 1Kazakh Research Institute of Livestock and Fodder Production, 51 Zhandosov Str., Almaty 050035, Kazakhstan; torehanov.aibyn@mail.ru (A.T.); nuralieva0602@gmail.com (U.N.); sheralieva95@mail.ru (Z.S.); 2Faculty of Geography and Environmental Sciences, Al-Farabi Kazakh National University, 71 Al-Farabi Ave., Almaty 050040, Kazakhstan; 3Department of Zoology and Animal Welfare, University of Agriculture in Krakow, 29 Listopada 56, 31-425 Krakow, Poland

**Keywords:** honey bees, native population, exotic population, geometric morphometry, introgression

## Abstract

*Apis mellifera pomonella* is one of two subspecies that represent the eastern limits of honey bee (*Apis mellifera*) distribution, and it is important to understand its biogeography and evolution. Despite this, *A. m. pomonella* was not investigated after its discovery 20 years ago. In particular, it is not known if it was hybridized or not with exotic subspecies introduced by beekeepers. In this study, we analysed the variation in honey bee forewing shape in Kazakhstan. Bees were collected from apiaries, where the origin of the queens was not controlled by beekeepers; they formed a group called “local bees”, and apiaries where queens declared as either *A. m. carnica* or *A. m. mellifera* were obtained from queen breeders. The two later groups were called “exotic bees”. We showed that local bees are still distinct from exotic ones. However, some samples showed signs of introgression with exotic subspecies from lineage C. In terms of wing shape, the local bees were most similar to lineage O. We concluded that the local bees most likely represented *A. m. pomonella*. We provided wing images and identification data, which can help to identify *A. m. pomonella* and protect it in the future. It is interesting that the nearby honey bee population sourced from China, which is not separated by any clear barrier to gene flow, belonged to lineage M.

## 1. Introduction

Originally, it was believed that the native distribution of the honey bee (*Apis mellifera*) covered Europe, Africa, and the Middle East [1]. Later, some native populations of this species were discovered further east in Kazakhstan [2] and China [3]. Despite the ecological and economic importance of honey bees, the eastern limits of their native distribution are poorly known. The reconstruction of these native limits is difficult because *Apis mellifera* was spread globally by humans [4]. The Tien Shan honey bee (*Apis mellifera pomonella* Sheppard & Meixner, 2003) is a subspecies of honey bee that was discovered 20 years ago in southern Kazakhstan [2]. It differs from other honey bee subspecies in both its morphology and mitochondrial DNA [2]. It belongs to the lineage O [5]. Its description was based on 31 colonies collected from three locations in the Tien Shan Mountains. It can be an important pollinator of wild apples (*Malus sieversii*), which occur in the same area. Its Latin name, *A. m. pomonella*, is related to Pomona, a Roman goddess believed to protect gardens. Since its discovery, no new data have been obtained about *A. m. pomonella*. In general, the knowledge about native and exotic honey bees in Kazakhstan is limited.

Kazakhstan is a large country extending west to east from the Caspian Sea to the Altai Mountains and north to south from the plains of western Siberia to the oases and deserts of central Asia. It has a continental climate, with hot summers and very cold winters. Its precipitation varies between arid and semi-arid, with the winter being particularly dry [6,7]. In a large part of the country, a lack of water prevents honey bees from naturally occurring. Before the introduction of beekeeping, honey bees probably occurred in Kazakhstan only in more humid areas in the north and south, but not in central Kazakhstan. In the north, honey bees occur in the Republic of Bashkortostan [8], the Altai Krai, and the Republic of Altai [9]. Those regions border Kazakhstan, and most likely honey bees occur naturally in the northern part of this country, at least in some more humid locations. Particularly suitable for honey bees are the mountain valleys in southern Kazakhstan, where the climate is milder and flowering plants are more abundant. The flowering season is longer there, as flowers can be present either on northern or southern slopes at various times of the year. In places less suitable for honey bees, they can survive with the help of beekeepers, and they continue to be introduced to the present territory of Kazakhstan [10]. After many years since the introduction of exotic bees and with scarce scientific records, it is not clear in which locations honey bees occur naturally.

Neither Ruttner [1] nor Crane [10] indicated that honey bees are native to Kazakhstan. It was reported that the migrants who arrived in Siberia after Yermak’s campaigns (in 1582) did not encounter bees there. In addition, Russian settlers who arrived in Kazakhstan around the same time did not find honey bees among the local residents [10,11]. However, those reports can only indicate a lack of beekeeping. The historical inhabitants of present-day Kazakhstan were mainly nomadic shepherds, and the transport of beehives was not practical for them. In this situation, beekeeping could be absent, even though wild honey bee colonies were present. In fact, there are some reports about honey hunting in the Altai Mountains [12,13], which suggest that wild honey bee colonies were present there. There is also some historical evidence confirming the presence of native honey bees in this region [14]. Ancient drawings depicting honey hunting were found near Teletskoye Lake [15,16], which is not far from the Kazakhstan border. Moreover, words related to bees, including honey, wax, and honeycomb, were present in the languages of the inhabitants of Altai, including Altaians and Jungars [11,17]. Both honey and mead (fermented honey) were used as food in this region [18]. Among the Teleutes people, who lived in northeastern Kazakhstan, there was a custom of drinking gold in vodka or mead at the time of taking an important oath [19]. Finally, there are historical reports about the presence of honey bees near Ust-Kamenogorsk from 1770 [20], when beekeeping was not present there.

Beekeeping, with the long-term maintenance and management of honey bees, started in Kazakhstan in 1786 [11]. After some initial failures, beehives were successfully introduced to the village of Bobrovka (eastern Kazakhstan) by a settler named Nikolai Fedorovich Arshenevsky. From there, honey beehives were introduced to the Almaty region in 1850, and later, from there, further west [21]. Until the beginning of the twentieth century, beekeeping was present only in the mountainous regions of southern Altai, Dzungarian Alatau, and western Tien Shan. Later, beekeeping penetrated into the lowlands of Kazakhstan. Earlier, it was restricted to the natural floodplains of the Ural, Syr-Darya, and Irtysh rivers. Only after the intensification of agriculture and the appearance of buckwheat (*Fagopyrum*) and sweet clover (*Melilotus*) crops did beekeeping become more widespread [21]. Bees are currently kept in all regions of Kazakhstan, except the southwestern region. There are about two hundred thousand honey bee colonies [22]. Along with traditional wooden beehives, modern plastic hives are also popular among beekeepers. Migratory beekeeping is more common than stationary beekeeping. There are historical records of importing bees from various places, including Bashkiria, Ukraine, and Kuban. This importation has intensified in recent years, and there are uncontrolled imports of queen bees. Some populations are isolated by deserts and mountains.

Within the wide distribution of the western honey bee, as mentioned earlier, there is considerable geographic variation in both its morphological and behavioural traits [1]. In order to describe the variation within this species, it was divided into four major evolutionary lineages and numerous subspecies [23]. The geographic variation of honey bees reflects their adaptation to the local environment, and they survive better in their local environment [24].

One of the methods that can be used for the investigation of the geographic variation of honey bees is wing geometric morphometrics. This method is based on landmarks indicated on wing images [25,26]. These landmarks can be determined manually [27] or automatically [28,29]. Wing geometric morphometrics has been successfully used for the identification of both species [30,31,32] and subspecies [33,34,35,36]. There are some reference samples [27,37] compatible with this method that can be used for comparisons with newly obtained data. 

The aim of this study was to use wing geometric morphometrics to describe the honey bees currently present in Kazakhstan. In particular, we were interested in confirmation that the native subspecies, *A. m. pomonella*, is still present there. To achieve this goal, bee samples were collected from different regions of Kazakhstan, including isolated locations where local, native bees may still be present.

## 2. Materials and Methods

### 2.1. Sampling

We used 1067 honey bee workers, which represented 71 colonies from 17 locations in Kazakhstan (Figure 1). In total, 29 of the colonies originated from apiaries in which beekeepers did not control the genetic background of their bees and did not obtain any queens from breeders. Those colonies formed a group called “local bees”. The other colonies originated from apiaries in which beekeepers used queens obtained from breeders. There were 38 colonies declared by breeders as *A. m. carnica* and 4 colonies declared as *A. m. mellifera*. Neither of these two subspecies occur naturally in Kazakhstan; therefore, the bees from the groups *A. m. carnica* and *A. m. mellifera* were called “exotic”. The samples were collected between 2020 and 2023 and stored in 96% ethanol. The right forewings of at least 15 worker bees from each colony were detached at their bases and secured with transparent adhesive tape on a piece of transparent plastic sheet (Appendix A). This method is used by some queen breeders and researchers in Russia [38]. All the wings from one colony mounted on a plastic sheet were scanned together using an office scanner, an Epson V600 PHOTO. The resolution of the wing image was 125,984 pixels per metre (3200 dpi). From the large image representing the whole colony, individual wings were cropped out and saved in separate files. Nineteen landmark coordinates were determined on each wing image using the IdentiFly (v. 1.8) [27]. The landmark positions were the same as those in Nawrocka et al. [27] (Figure 2). The dataset used in this study, including forewing images, landmark coordinates, and the geographic coordinates of the sampling locations, is available on the Zenodo website [39]. The preparations with mounted wings are stored for future reference at the Kazakh Research Institute of Livestock and Fodder Production, Almaty, Kazakhstan.

### 2.2. Statistical Analysis

All the analyses were conducted in R (v. 4.0.3) [40] using RStudio (v. 2022.12.0). The whole statistical analysis is available as an R script (Appendix A). The cubital index was calculated as the ratio of the two forewing veins [41]. It was compared between the groups using an analysis of variance (ANOVA). The landmark configurations were aligned using a generalised Procrustes analysis in the geomorph package [42]. The aligned coordinates were averaged within the colony, and all the subsequent calculations were based on the colony averages. In order to reduce the number of dimensions and visualise the variation present in the samples, a principal component analysis (PCA) was used. The subsequent analysis was based on the first 34 principal components. The differences in wing shape between the groups were analysed with a multivariate analysis of variance. The discrimination between the groups was analysed using a canonical variate analysis (CVA), which is also called a linear discriminant analysis (LDA), calculated using the package MASS [43]. The samples were classified as evolutionary lineages using data provided by Nawrocka et al. [27].

## 3. Results

The cubital indexes (±SD) for the local bees, *A. m. carnica*, and *A. m. mellifera* were: 2.30 ± 0.22, 2.69 ± 0.22, and 1.72 ± 0.12, respectively. The cubital index differed significantly between the three groups (analysis of variance: F = 52.3, *p* < 10^−13^). The principal component analysis of the wing shape revealed that the honey bee colonies from Kazakhstan formed a single cluster, but the three groups (local bees, *A. m. carnica*, and *A. m. mellifera*) were separated to some degree along the first principal component, which explained 33.9% of the variance (Figure 3). The three groups differed significantly in wing shape (multivariate analysis of variance: F = 19.0, *p* < 10^−15^).

The three groups were clearly separated by the canonical variate analysis (Figure 4), and the colonies were classified without errors into their groups using leave-one-out cross-validation. The local bees from southern Kazakhstan were more similar to *A. m. carnica* than to *A. m. mellifera* (Table 1).

The classification into lineages based on the data from Nawrocka et al. [27] showed that the colonies of local bees were classified as lineage O (12 colonies), lineage C (11 colonies), and lineage A (6 colonies); all the *A. m. carnica* colonies were classified as lineage C; and the *A. m. mellifera* colonies were classified as lineage M (2 colonies) and lineage A (2 colonies) (Figure 5).

When the colonies of local bees were considered as a separate group, they were discriminated without errors from all four honey bee lineages (Figure 6). The local bees were most similar to lineage O (Table 2).

## 4. Discussion

The data presented here showed that the local bees in southern Kazakhstan form a distinct group that is different from the exotic bees in this country (Figure 4). The cubital index of the local bees (2.30) was within the confidence interval of the value 2.24 ± 0.20 (mean ± SD) provided for *A. m. pomonella* [2]. In the original description, based on mitochondrial DNA, *A. m. pomonella* was considered to belong to mitochondrial lineage C, which includes both C and O, but later, it was considered to belong to lineage O [5]. This later assignment is consistent with the geographical proximity of south Kazakhstan to Iran, where lineage O is present [44]. The local honey bees in southern Kazakhstan can most likely be considered to be *A. m. pomonella*, which has been described in this area [2]. In this study, the wing shape of the local bees was most similar to lineage O (Figure 6, Table 2), which is consistent with the earlier findings. We found *A. m. pomonella* in two new locations not reported before (Koksu, Almaty region, and Karakonyz, Zhambyl region), but were not able to confirm its presence in Kurmetty in the Almaty region, which was reported in an earlier study [2] (Figure 1). According to the local inhabitants of Kurmetty, in recent years, there have been no stationary apiaries present in this location; however, during the flowering season, migratory apiaries can occasionally be present there.

Although the local bees formed a distinct group and were most similar to lineage O, 11 out of 29 colonies (37.9%) were classified as lineage C. This could be an effect of introgression with *A. m. carnica*, which was introduced to all regions of Kazakhstan, including the natural range of *A. m. pomonella* (Figure 1). Moreover, six colonies of local bees (20.7%) were classified as lineage A. This could be related to the importation of hybrids called Buckfast, which have some African ancestors [45]. It is also possible that these six colonies were hybrids between lineages O, C, and M, as samples with intermediate phenotypes tend to be classified as lineage A [37]. The importation of exotic bees to Kazakhstan started in historic times and intensified recently (for details, see the introduction). Because beekeepers do not control the mating between queens and drones, introduced bees hybridise with local ones [46]. This can lead to the loss of some genetic variation and, in the case of mass importation, the extinction of *A. m. pomonella*. The local bees described here deserve to be protected from further introgression with exotic bees. In order to achieve this, the region of the Tien Shan Mountains should be designated as a conservation area. Prohibition of the importation of exotic honey bee subspecies into Kazakhstan would benefit this conservation. If the breeding of non-native bees continues, it should not be performed in the native honey bee range. In a country as large as Kazakhstan, isolated locations in the central part of the country can be used for this purpose.

*A. m. pomonella* is a unique subspecies at the eastern limits of honey bee distribution. Further east, there is only *A. m. sinisxinyuan* in China [3]. It is interesting that *A. m. pomonella* belongs to lineage O and *A. m. sinisxinyuan* belongs to lineage M. The Chinese population is on the western side of the Tien Shan Mountains, and it is only about 300 km from Tuyuk in Kazakhstan, where *A. m. pomonella* was detected in both the previous [2] and this study. Between the two locations, there is no clear geographic barrier preventing gene flow (Figure 1). It remains to be explained why the two neighbouring honey bee populations belong to different lineages. One explanation could be the patchiness of honey bee distribution in alpine and arid climates. It is possible that the distribution of *A. m. sinisxinyuan* is restricted to one valley surrounded by high mountains and arid areas [3,47]. Another explanation could be human intervention. If the two populations are maintained by beekeepers, the country border between Kazakhstan and China can restrict migratory beekeeping and the exchange of genetic material. In order to better understand the variation of honey bees in this region, it is important to broaden the sampling to include a larger area of Kazakhstan. Such an investigation should describe the whole distribution of local bees and their geographic variation. There should be included apiaries where queens were not obtained from queen breeders, or workers could be collected from flowers to obtain samples from a larger number of surrounding colonies, both managed and feral. In northern Kazakhstan, there can be expected a higher similarity to lineage M, which was recorded in the Republic of Bashkortostan [8] and the Altai Republic [9] close to the Kazakhstan border. On the other hand, in southern Kazakhstan, there can be expected a higher similarity to lineage O. It would be interesting to know how the transition between these two lineages formed in areas where honey bees did not occur naturally.

We were not able to obtain wing images or any other comparative material, apart from the publication with a description of *A. m. pomonella* [2], to verify our results. We provide wing images [39] which can be used as reference material for future studies. We also provide data saved in a DW.XML file (Appendix A), which can be easily used for the identification of an unknown sample using the IdentiFly (v. 1.8) software [27]. Apart from the data provided in this study, there are known mitochondrial DNA sequences [2]. These sequences can be used to confirm, with a greater certainty, that the local bees are *A. m. pomonella*.

## 5. Conclusions

It can be concluded that *A. m. pomonella* is still present in southern Kazakhstan. The local bees in the Tien Shan Mountains require urgent protection, because there are some signs of introgression with exotic bees. The wing images provided in this study can help to identify and protect *A. m. pomonella*.

## Figures and Tables

**Figure 1 life-13-01860-f001:**
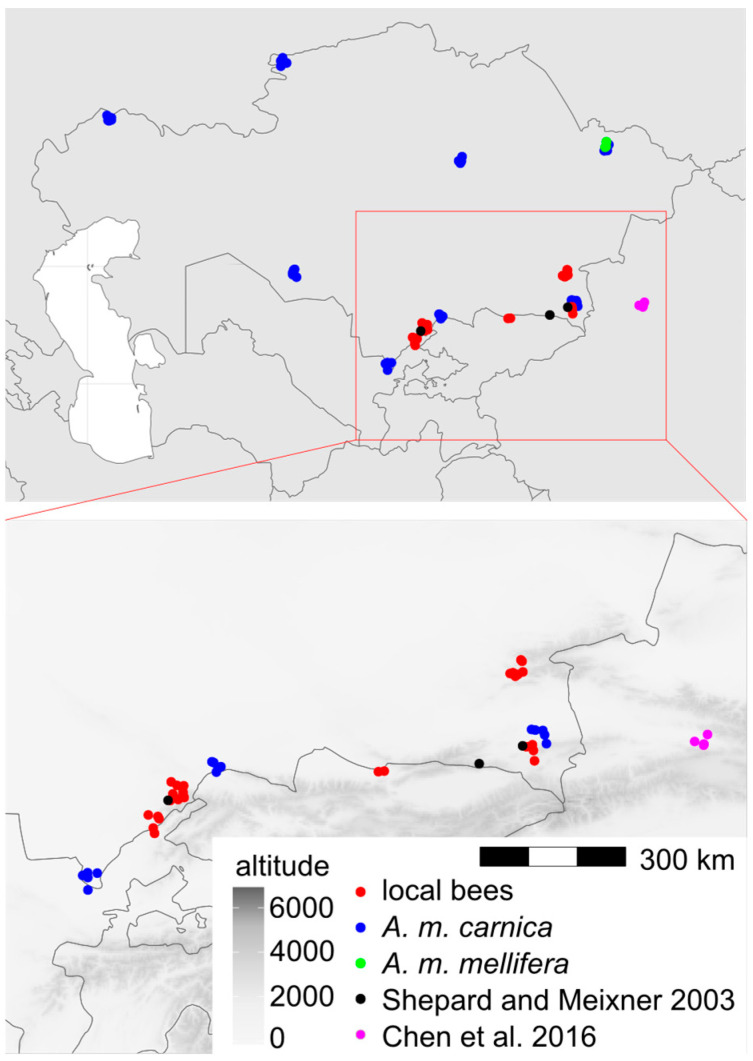
Map of Kazakhstan with marked locations from which samples were obtained. Jitter was used to show multiple samples from the same location. Black and magenta markers indicate locations investigated in earlier studies: refs. [2] and [3], respectively.

**Figure 2 life-13-01860-f002:**
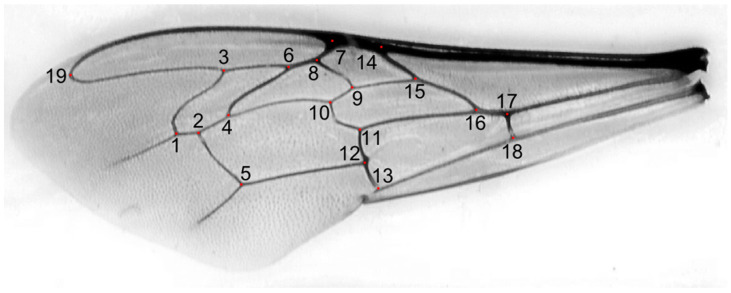
Forewing of a honey bee worker from southern Kazakhstan. Red dots indicate 19 numbered landmarks used for wing measurements.

**Figure 3 life-13-01860-f003:**
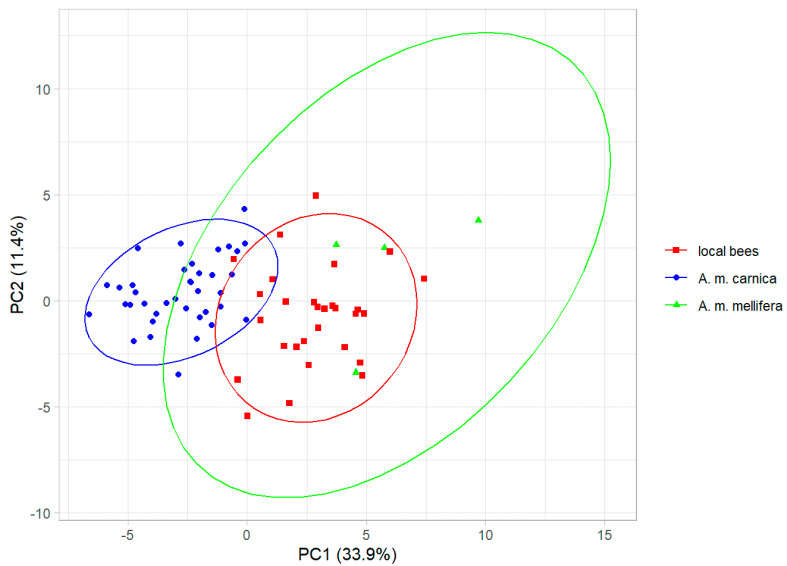
The first two principal components of wing shape. Ellipses indicate 95% confidence regions.

**Figure 4 life-13-01860-f004:**
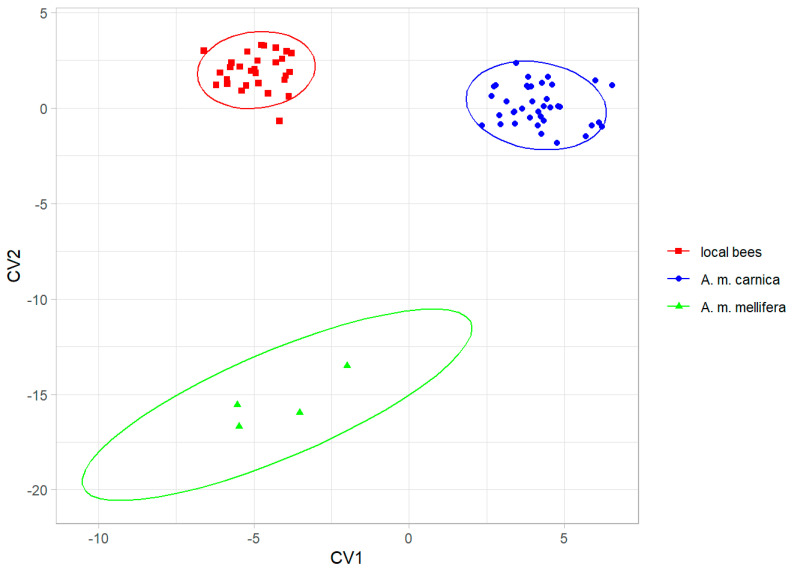
Discrimination between local bees, *A. m. carnica*, and *A. m. mellifera*, based on the first two canonical variates. Ellipses indicate 95% confidence regions.

**Figure 5 life-13-01860-f005:**
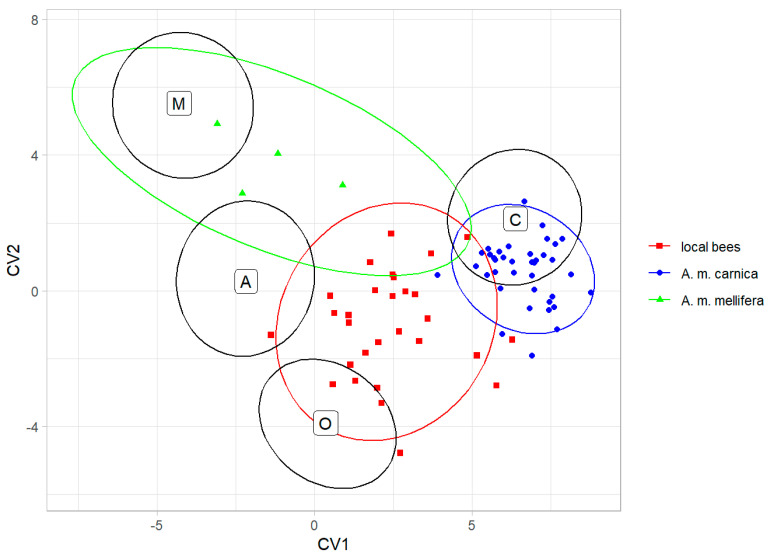
Classification of colonies from Kazakhstan as honey bee lineages using data provided by Nawrocka et al. [27]. Ellipses indicate 95% confidence regions. Black letters and ellipses refer to the lineages, and colour points and ellipses refer to the experimental groups.

**Figure 6 life-13-01860-f006:**
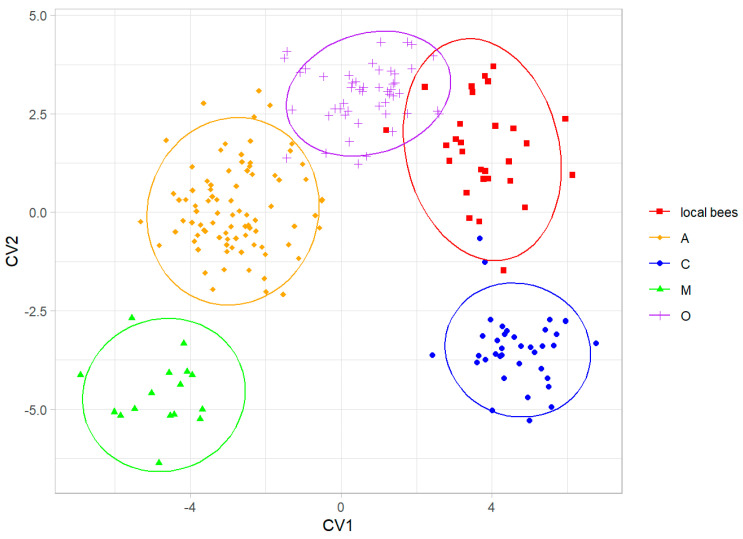
Discrimination between colonies of local bees and honey bee lineages using the first two canonical variates. Ellipses indicate 95% confidence regions.

**Table 1 life-13-01860-t001:** Differences in wing shape between local bees, *A. m. carnica*, and *A. m. mellifera*. Mahalanobis distances between groups are in the upper triangle. The significance values of pairwise comparisons between groups are in the lower triangle.

	Local Bees	*A. m. carnica*	*A. m. mellifera*
Local bees	-	9.269	17.385
*A. m. carnica*	10^−4^	-	17.643
*A. m. mellifera*	10^−4^	10^−4^	-

**Table 2 life-13-01860-t002:** Differences in wing shape between local bees and evolutionary lineages. Mahalanobis distances between groups are in the upper triangle. The significance values of pairwise comparisons between groups are in the lower triangle.

	Local Bees	A	C	M	O
Local bees	-	7.960	7.562	11.886	7.207
A	10^−4^	-	8.513	8.061	6.091
C	10^−4^	10^−4^	-	11.155	8.008
M	10^−4^	10^−4^	10^−4^	-	10.209
O	10^−4^	10^−4^	10^−4^	10^−4^	-

## Data Availability

The dataset from Kazakhstan, including the wing images, landmark coordinates, and geographic coordinates of sampling locations, is openly available in Zenodo at [https://doi.org/10.5281/zenodo.8128010] (accessed on 17 August 2023). The statistical analysis is openly available as an R markdown document in WorkflowHub at [https://doi.org/10.48546/WORKFLOWHUB.WORKFLOW.559.1] (accessed on 28 August 2023).

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
