# Peer review of "In Search of Apis mellifera pomonella in Kazakhstan"

_life, 2023, doi:10.3390/life13091860_

Round 1
Reviewer 1 Report
That is quite an interesting paper in various aspects including phylogeography, beekeeping, genetic resources, and methodology, and can be read by a wide range of interests, so the author should slightly revise the manuscript for a better description of what they did.
Abstract needs clarifications (see comments inside)
The introduction is OK, but there was a gap in the statements. The authors immediately jumped to the methodology of how to find variation, but have to describe the importance and significance of variations, first.
The results is straightforward, but without details, as many items referred to https://zenodo.org/record/8128010
The discussion was strong and successful. I put a small but important comment to improve it.
References are OK
Best Regards

very small points are commented on.
Author Response
Thank you for the valuable comments from Reviewer 1.
We have corrected the manuscript according to most of the Reviewer 1 comments. In particular, we have added some information about honey bee variation in lines 96–101.
We agree that the information about local and exotic bees was incomplete. We have added more information in lines 118–124 and in the abstract.
We have changed "non-native" to "exotic" in the whole text.
We have added more information about wing mounting in lines 125–128 and 136–138. Figure 1S shows the mounted wings.
We have included most of the minor corrections indicated in the pdf file except for the comment in the results: "Provide reference for this categorization of the lineages", which was not clear to us.
Reviewer 2 Report
Dear Authors,
Thank you for submitting your manuscript, which presents an insightful investigation into honeybee populations in southern Kazakhstan. The manuscript is an important contribution to the field, shedding light on the morphological differences among various bee subspecies and emphasizing the need for conservation.
In my review, I have focused on the following key areas:
Verification of Results: Comparison against physical specimens and rephrasing suggestions.
Scope and Generalizability: Exploration of study limitations and suggestions for broader research.
Historical and Geographical Context: Enhancing the background information to emphasize connections to the region.
Conservation Aspects: Recommendations for clearer articulation of conservation needs and strategies.
Methodological Concerns: Specific comments on methods and presentation.
The manuscript under review presents an investigation into the honeybee populations in southern Kazakhstan, specifically focusing on local bees and imported stocks of the subspecies A. m. carnica and A. m. mellifera via wing morphology. The authors identify distinct morphological differences among these groups. The findings reveal that the local bees most likely correspond to A. m. pomonella, with evidence of introgression with non-native honeybee subspecies. The research underscores the need for protective measures for the local honeybee populations in Kazakhstan and a ban of importation of non-local bees. Moreover, the study contributes valuable reference material for future studies in this region. I think the study is well suited for the journal and methods are appropriate. However, I have some comments on the manuscript.
As far as I understand the authors were not able to obtain wing images or any other comparative material apart from the cubital index in the publication that originally described A. m. pomonella. This means they were not able to directly verify their results against physical specimens. Therefore, I think it is better to rephrase these two sentences: “They can be considered A. m. pomonella, which was described in this area [2]” (line 179) and “We conclude that the local bees represent A. m. pomonella.” (line 18). I suggest using in both cases “most likely”.
The study appears to be limited to mainly southern Kazakhstan, which could limit the generalizability of its findings to a broader bee population. Please discuss this. It would be interesting to see the wing morphology results of foraging honeybees collected all over the country on flowers (across a grid). I suggest that the authors consider this as a follow-up study.
The introduction nicely delivers the history of beekeeping in Kazakhstan. I miss the connection to the species name of A. m. pomonella. Malus sieversii, the wild ancestor of the domesticated apple (Malus domestica), is native to the region that includes the Tien Shan mountains of Kazakhstan (see the Discussion of Sheppard & Meixner 2003). In regions where A. m. pomonella is native, it is conceivable that they might have played (and still play) a significant role in the pollination of local apple orchards. Therefore, I would like you to more clearly state this in the introduction that wild populations were probably present in some parts of Kazakhstan.
You also mention that imports from Bashkiria (and Ukraine and Kuban) historically occurred. Bashkiria still has wild A. m. mellifera populations (in the Schulgan-Tasch national park) and I recommend mentioning them (see Ilyasov 2015). As this area is quite near to the border of Kazakhstan they might also be affected if colonies of C-lineage are imported to Kazakhstan.
Regarding conservation of the local bee subspecies:
Please rephrase this: “Kazakhstan is a large country, and for breeding non-native bees, there should be regions outside of the native honey bee range.” (line 207) as it is for the reader maybe not immediately clear what you mean. I suggest something like this: Kazakhstan is a large country. Therefore, breeding of non-native bees should not be done in the native honey bee range. Furthermore, I would like the authors to make a clearer point on the problematic importation of non-local queens. This should be banned.
I think it would be good if you mention the ecology of the different subspecies and why it is important to conserve them (e.g. cite Büchler 2014 on the influence of genetic origin).
Some more specific comments regarding the methods sections:
Methods:
You state: “The right forewings of at least 15 worker bees from each colony were detached at their bases and secured with transparent adhesive tape on a piece of transparent plastic sheet.“ (line 104). Please show an example/photo in the main text or the supplementary information.
“The resolution of the wing image was 125984 pixels per metre (3200 dots per inch)” (line 107). I think dpi would be easier to understand.
Please consider changing the colour choice of Fig. 1. Light green and red is used also in the elevation gradient colour. Also grey (and black) are not easy to locate on the map.
Thanks for this interesting study.
Author Response
Thank you for the valuable comments from Reviewer 2.
We agree that there is some uncertainty about the identification of local bees as A. m. pomonella. We have indicated this as suggested in lines 20, 215. In lines 270–272 we added information that an investigation of mtDNA should be more conclusive.
We agree that local bees should be investigated over a larger area. We included this information in lines 253-264. We plan to conduct such an investigation in the future.
We have added more information about the link between A. m. pomonella and apple orchards in lines 37–40.
We have added information about honey bees living close to the northern border of Kazakhstan in lines 49–52 and discussed this problem in lines 259–261.
We agree that a ban on the import of exotic honey bees would be beneficial for the conservation of local bees. We have included this information in lines 237–238.
We have added information about the link between geographical variation and survival in lines 99–101.
We have added more information about wing mounting in lines 125–128 and 136–138. Figure 1S shows the mounted wings.
We prefer metric units, but we also provide resolution in dpi. We replaced "dots per inch" with "dpi".
In order to make colour markers better visible, we have used grayscale to indicate altitude in figure 1.
Reviewer 3 Report
Using morphometry, you proved the existence of local bees Apis mellifera pomonella. Fifteen wings per sample is a smaller number than the number of drones with which the queen mated, but it is still vital proof of the existence of local breeds. An assessment of the danger to the local species from the introduced breeds would contribute to the importance of the results.
Author Response
We agree with Reviewer 3 that measurements of 15 wings per colony will not describe the whole morphological variation related to all queens partners. However, this was not our aim. Our focus was not on a particular queen and all her partners but on a more general genetic background. We believe that our imperfect assessment adequately describes the wing phenotype of honey bees from a particular colony. In most other studies, the genetic background of a colony is assessed using a smaller number of workers. In molecular studies, there is often only one worker per colony investigated.
Round 2
Reviewer 2 Report
Thanks for the revised version. Just one last comment: I think the term feral is not correct. Feral refers to wild-living colonies that have recently escaped management or, analogously, to populations of wild-living colonies that are not self-sustaining. In line 65 and 67 "wild" would be more correct if it is assumed that honeybees are native to parts of the country (e.g. " beekeeping could be absent even though wild honey bee colonies were present"). In line 259 one could write "unmanaged" or "wild-living" as it is (probably) not clear if colonies are present and if they are self-sustaining.